# Synthesis of Fluorogenic Arylureas and Amides and Their Interaction with Amines: A Competition between Turn-on Fluorescence and Organic Radicals on the Way to a Smart Label for Fish Freshness

**DOI:** 10.3390/molecules26051404

**Published:** 2021-03-05

**Authors:** Javier García-Tojal, José V. Cuevas, María-Josefa Rojo, Borja Díaz de Greñu, Carla Hernando-Muñoz, José García-Calvo, Mateo M. Salgado, Tomás Torroba

**Affiliations:** Department of Chemistry, Faculty of Science, University of Burgos, 09001 Burgos, Spain; qipgatoj@ubu.es (J.G.-T.); jvcv@ubu.es (J.V.C.); mjrocam@ubu.es (M.-J.R.); bdpuertas@ubu.es (B.D.d.G.); chm1001@alu.ubu.es (C.H.-M.); jgcalvo@ubu.es (J.G.-C.); mmsalgado@ubu.es (M.M.S.)

**Keywords:** arylureas, turn-on fluorescence, organic radicals, smart label, volatile amines

## Abstract

We describe the synthesis of fluorogenic arylureas and amides and their interaction with primary or secondary amines under air and light in organic-aqueous mixtures to give rise to a new class of persistent organic radicals, described on the basis of their electron paramagnetic resonance (EPR), as well as UV–vis, fluorescence, NMR, and quantum mechanics calculations, and their prospective use as multi-signal reporters in a smart label for fish freshness.

## 1. Introduction

Persistent organic radicals are of immense importance for the preparation of new hybrid materials with spin-probe and spin-label functions [1], magnetic materials [2], electrical transport [3], molecular spintronics [4], and surface-confined electroactive molecules for single-molecule switching devices [5], and they remain integral in classical areas such as catalysis for the oxidation of organic molecules [6] and DNA damage [7]. Persistent radical cations [8] and radical anions [9], examples of organic mixed-valence compounds [10], are also important, along with their neutral organic radical parents, for the discovery of new organic materials and the development of improved devices. Amines and their derivatives are extensively used as chemicals in abuse drugs, cosmetics, polymers, etc. [11]. They are also important markers in food industries and medical diagnosis, as they are involved in many biological processes, including as neurotransmitters [12]. The detection and discrimination of amines are important issues in environmental protection and human health. Common protocols for amine analyses are based on chromatographic methods [13], which are quite time-consuming; therefore, the development of fluorescent sensors for amines has become a desirable alternative given their simple instrumentation and fast response time. Although fluorescent sensors lack selectivity in some applications [14,15,16,17,18], the use of multiple-signal sensors can remediate this adverse selectivity. Herein, we present an unusual family of new fluorogenic aminoaryl-indane-based probes that are also able to form persistent radical species in the presence of primary or secondary amines, and we describe their prospective use as multiple-signal fluorogenic reporters for the detection of volatile amines.

## 2. Results and Discussion

The synthesis of arylureas used in this study was performed using an easy Suzuki reaction between 5-bromoindanone and 4-aminobenzeneboronic acid pinacol ester, followed by a Knoevenagel reaction with malononitrile and a reaction with selected mono- and bis-isocyanates (Scheme 1). To compare their fluorogenic properties, selected bis-amides were also prepared in a straightforward reaction of the starting aminoindane derivative and acid chlorides (Scheme 1).

In this way, chemical probes **1**–**7** were prepared (Figure 1). Probe **1** was previously reported as an excellent fluorogenic sensor for the determination of 3,4-methylenedioxymethamphetamine (MDMA) from ecstasy tablets [19]. The interaction of primary and secondary amines with the fluorogenic probe was the key step in their selective discrimination from tertiary amines, anilines, and heterocyclic amines. We observed that the presence of primary or secondary biogenic amines initially promoted a sudden fluorescence in the probe, which was followed by gradual emission extinction.

The kinetics of the process was specific for every reported amine, but the reason for the emission extinction was unclear until we noticed that it was caused by the presence of radical species generated after the probe/amine interaction. We selected a typical amine derivative, pyrrolidine, and studied the interaction of pyrrolidine and **1** using titration experiments monitored by electron paramagnetic resonance (EPR), as well as by NMR, UV–vis and fluorescence spectroscopy, in order to understand the type of interaction involved in the generation of the radical species. First, the effect of the addition of pyrrolidine to an equimolecular solution of bis-diarylurea **1** in dimethylsulfoxide (DMSO) was studied by X-band EPR measurements at room temperature under ambient light and air. Time-dependent absorption was detected after faint discoloration of the dark orange solution of **1**. The signal was sensitive to the solvents used. The low solubility of **1** compelled us to use DMSO, the only common solvent in which this urea is moderately soluble, but a continuous decrease in the intensity of the EPR signal from minute 5 was observed if DMSO was used as the only solvent. However, the radical life-time increased with the addition of water; therefore, aqueous mixtures were used for the measurements. In some of the experiments, acetone was added in an attempt to improve the resolution by decreasing the viscosity of the solvent mixture, but no highly significant improvement was attained. As a typical example, Figure 2 shows the evolution of the EPR spectra over time for a mixture of **1** and pyrrolidine at room temperature. The measurements were performed on a mixture prepared from 100 μL of 10^−2^ M stock solution of **1** in DMSO diluted with 100 μL of water, 100 μL of acetone, and 100 μL of DMSO. The mixture was stirred for 1 h, followed by the addition of 100 μL of a 10^−2^ M solution of pyrrolidine in DMSO. After shaking, 200 μL of the non-deoxygenated mixture was pipetted into a flat cell. A spectrum with 18 equally spaced lines was registered at room temperature, and its intensity increased for the first 11–12 min and then decreased to disappearance within the next 40 h (Figure 2).

Once the experimental conditions were optimized, the best protocol consisted in preparing an orange mixture containing 100 μL of 10^−2^ M stock solution of **1** in DMSO, which was diluted by adding 100 μL of water and 200 μL of DMSO and stirred for 2 h, followed by the addition of 100 μL of a 10^−2^ M solution of pyrrolidine in DMSO, at which moment the color of the mixture slightly clarified. After shaking, 200 μL of the final solution (2 × 10^−3^ M in DMSO/water 4:1 *v*/*v*) was pipetted into a flat cell, where the measurements started 11 min after the addition of the amine. The obtained spectrum is shown in Figure 3. The best fit (correlation coefficient R = 0.9930) was reached when taking into account the hyperfine coupling of the unpaired electron to two non-equivalent ^14^N nitrogen nuclei (I = 1) and three sets of non-equivalent ^1^H hydrogen atoms (I = ½). The parameters arising from the fit were *g* = 2.0035, *A*_N1_ = 0.180 mT (1N, 1.68 10^−4^ cm^−1^), *A*_N2_ = 0.060 mT (1N, 0.56 × 10^−4^ cm^−1^), *A*_H1_ = 0.260 mT (1H, 2.43 × 10^−4^ cm^−1^), *A*_H2_ = 0.120 mT (2H, 1.12 × 10^−4^ cm^−1^), *A*_H3_ = 0.060 mT (2H, 0.56 × 10^−4^ cm^−1^), and line-width ∆*H* = 0.055 mT.

The use of *n*-butylamine instead of pyrrolidine yielded a similar spectrum. A comparison between the EPR spectra of **1** after the addition of pyrrolidine or *n*-butylamine is given in Figure 4; the experimental conditions were the same as those reported for previous pyrrolidine experiments, but the microwave frequency in the case of *n*-butylamine was 9.7750 GHz. An equimolecular solution of **1** and DABCO in similar conditions gave rise to a very weak isotropic signal with a non-resolved hyperfine structure (g ≈ 2.0035, ∆*H* = 0.500 mT (Appendix A). In the case of the interaction of **1** with tetrabutylammonium cyanide, signals weaker than those obtained from **1** and pyrrolidine were registered, but they showed the same hyperfine pattern (Appendix A). On the contrary, a very weak and unresolved initial signal was obtained when aqueous NaOH (1 M) solution was added (Appendix A). After one day, this signal evolved to an intense isotropic Gaussian-type signal (*g* = 2.0029, ∆*H* = 0.580 mT), and the color of the solution became dark green (Appendix A). The observed radical species are loosely related to stable radicals formed from the decomposition of azobisisobutyronitrile (AIBN) [20,21], in which stabilization by a capto-dative effect may be involved [22,23,24,25].

The EPR results proceeded in parallel to the reported fluorescent results for **1**, which was only sensitive to the presence of primary and secondary amines and generated an intense fluorescent signal, but it was not sensitive to the presence of tertiary amines, with the exception of a weak sensitivity to DABCO [19]. We then recorded the EPR spectra of bis(aryl-benzyl)urea **2** (a fluorescent molecular ruler for ω-amino acids) [26], bis-arylureas **3**–**5**, and diamides **6** and **7** (the central diamide linker was used in fluorescent polyamide chemical probes) [27] in the presence of 1 equiv. of pyrrolidine in the same conditions. In spite of their low intensities and deficient resolutions, the measurements performed on those bis- and mono-ureas suggested the attainment of the same EPR signal with different intensities, with the most intense signals related to bis-urea systems in the order **1** >> **2** > **3** ≈ **4** ≈ **5**, indicating that bis-aryl-urea **1** produced a more intense and better resolved signal than bis-benzylurea **2**, and both produced more intense signals than mono-arylureas **3**–**5**. However, no signals in the EPR spectra were obtained in the experiments with bis-amides **6**–**7** under the same experimental conditions; therefore, they were not further studied (Figure 5). 

EPR intensity changes were accompanied by changes in the color of the solution, from orange to pale yellow, after the addition of the amine to **1**–**5** solutions. Therefore, we performed UV–vis and fluorescence titrations with **1** and pyrrolidine as a representative case under the same conditions, generating changes in color and an intense fluorescent signal in either mixed solvents, DMSO/water/acetone, or pure DMSO, which was followed by kinetic measurements (see Appendix A). The addition of up to several equivalents of pyrrolidine in DMSO to a solution of **1** (5.0 × 10^−5^ M) in DMSO at room temperature caused the appearance of a UV–vis absorption band centered at 305 nm and the disappearance of the absorption band centered at 405 nm (Appendix A). In the interval between 0 and 1 equiv. of the amine, an isosbestic point at 336 nm was observed. During the titration experiment, we did not observe bands in the 650–900 nm region of the absorption spectrum, which are characteristics of the formation of radical anion species [28,29], so their presence was excluded. UV–vis titration experiments in a DMSO/water 4:1 (*v*/*v*) mixture produced similar results. We calculated the binding constants from UV–vis and fluorescence titrations (Appendix A), which were in agreement with those previously reported for biogenic amines [19]. Therefore, the behavior of **1** in the presence of pyrrolidine was similar to that observed for related amines, so we assumed that the radical species was formed in the presence of all reported primary or secondary amines. 

The evolution of the radical species was followed by ^1^H and ^13^C NMR spectroscopy of equimolecular solutions (2.5 × 10^−3^ M) of **1** and pyrrolidine in deuterated DMSO. The ^1^H NMR spectrum of **1** is shown in Figure 6 in blue. The purple spectrum in Figure 6 was recorded immediately after the addition of pyrrolidine, which shows upfield displacements in one of the NH peaks near δ 9.0, the closest aromatic proton signal at δ 8.25 (the aromatic proton signals between δ 7.5 and 8.0 also underwent upfield changes), and downfield displacement of one of the methylene groups at δ 3.2, indicated by the blue and purple arrows, but the resolution of the spectrum did not change, indicating only hydrogen bond interactions. The green spectrum of the same sample in Figure 6 was recorded after 8 h, in which the NH protons produced a broad signal near 9 ppm, all aromatic protons between δ 7.3 and 8.3 produced a large unresolved signal between δ 8 and 7.6, and the methylene signals appeared as large multiplets between δ 3 and 3.5, indicating a strong perturbation of the resolution (green arrows). The signals at δ 7.5 and 7 did not undergo large changes. 

The ^13^C NMR spectrum of **1** is shown in Figure 7 in blue. The purple spectrum in Figure 7 was recorded 8 h after the addition of pyrrolidine, which shows the disappearance of the signals at δ 180 and 70 (corresponding to the exocyclic double bond), of the two peaks at around δ 114 (corresponding to the two cyano groups), and of the methylene signal near δ 30. These changes are indicated by the blue and purple arrows. The spectrum also shows the appearance of a new cyano group signal at δ 109, indicated by the red arrow. The signal near δ 155 diminished, and the rest of the aromatic carbon signals between δ 150 and 123 experienced several changes, but the signals between δ 120 and 118 remained unchanged. Apparently, the bis-aryl ether moiety was not affected by the presence of the radical species in either ^1^H NMR or ^13^C NMR spectra, and all changes were in the arylindane moiety. The most important changes affected the exocyclic dicyanovinyl group and the closest methylene group, indicating the main positions where the radical species was localized. The signals corresponding to the pyrrolidine group remained unchanged in all spectra.

We hypothesized that the addition of an electron to the cyanovinyl moiety produced a weakening of the triple bond that thus accumulated negative charge on the nitrogen, which acquired a more C=N character, a structure that can be stabilized by the capture of hydrogen, or a C=C=N··H group in the presence of water. Both IR and NMR data pointed to the weakening of the C≡N bond, in agreement with this hypothesis.

A structure featuring all characteristics was subjected to quantum chemical calculations, as shown in Figure 8. The geometric structure of **[1]**^•^ was optimized by using the Becke three-parameter functional [30] and the Lee–Yang–Parr correlation functional [31] (B3LYP) at the 6-31G* basis set for all the elements, implemented in the Gaussian 03 (Revision C.02) program suite [32]. The unrestricted protocol was used for open-shell systems. From the energy-optimized structures, the singly occupied molecular orbital (SOMO) is represented in Figure 7. The SOMO is centered on the dicyanovinyl moiety and displays a π-antibonding character located over the C=N bond of the protonated nitrogen and on the 1-indan carbon atom and the aryl group, as well as π-bonding over the indane aryl group. The molecular orbital has slight contributions from σ-bonding interactions between the hydrogen atom and the ketenimine nitrogen atom, as well as the hydrogen atoms from the alpha-methylene carbon atom of the five-membered ring. The large electronic delocalization seems to be responsible for the high stability and long life of the radical. SOMO + 1 was located on the arm of the molecule that was not affected by the formation of the radical. It mainly has a π-antibonding character located over the aryl fragment near the nitrile groups and a π-antibonding character in the nitrile groups. Nevertheless, this orbital has a slight π-bonding character in the C–C interactions of the nitrile groups and the 1-indane carbon atom with the aryl fragment. SOMO + 1 was 2.03 eV (195.65 kJ mol^−1^, 46.76 kcal mol^−1^) higher in energy than SOMO. SOMO − 1 is a π-bonding molecular orbital delocalized over the phenyl rings directly bonded to the ether oxygen atom and with slight π-antibonding interactions with a p orbital of this oxygen atom and with p orbitals of the urea nitrogen atoms. SOMO − 1 was 1.03 eV (99.24 kJ mol^−1^, 23.72 kcal mol^−1^) lower in energy than SOMO. The energy gap between SOMO and SOMO + 1 was within the visible light region, 1.55–3.26 eV (150–315 kJ mol^−1^, 35.85–75.29 kcal mol^−1^); therefore, light excitation and proton transfer may delocalize the radical between the two arylindane moieties. This fact explains why in ^1^H/^13^C NMR, both arylindane groups seem to be affected to the same extent by the formation of the radical species. The interaction must be intramolecular. Indeed, a mixture of **1**, pyrrolidine, and 5-(4-aminophenyl)-1-dicyanomethyleneindane (the amine precursor of **1**) in deuterated DMSO showed vinyl signals at δ 180 and 70 after an 8-hour ^13^C NMR recording. The fact that SOMO − 1 was localized exclusively at the diarylether moiety explains the fact that this group appeared unchanged in all spectroscopic experiments. Although the diarylether group seems to have little influence on the stability of the radical species (there is no electronic interaction), this link seems to be optimal for electron delocalization between the two indane moieties. An electronically isolated link such as the *m*-xylyl group in **2** produced a much weaker radical species in similar conditions, and none of the conjugated bis-amide links, such as those in **6**–**7**, produced radical species in the presence of pyrrolidines.

From previous studies and the new experiments, we assumed that compound **1** was a non-fluorescent compound, in contrast to fluorescent compound **2**, because of a charge transfer from the diarylether moiety to the fluorescent arylindane groups. The addition of primary or secondary amines such as pyrrolidine or *n*-butylamine inhibited the charge transfer by forming hydrogen bonds, thus giving rise to a fluorescent intermediate that underwent electron transfer [33] from the amine group to the indane moiety, facilitated by simultaneous proton transfer from water [34], giving rise to radical species characterized using several spectroscopic techniques (Figure 9). A quick inspection of the evolution of fluorescence (left) and paramagnetic signals (EPR, right) in Figure 9 clearly shows that the extinction of fluorescence evolved in parallel to the increase in the EPR signal, so the asymptotic minimum fluorescence was reached at roughly the same time as the maximum EPR signal. We only observed the stable radical species [**1**]^•^, and therefore, the radical species that could form from the amine were reversed in the presence of water and air compared to the neutral amine, which appeared unchanged in all NMR experiments.

These chemical probes are the first examples of multi-signal reporters for the discrimination between primary, secondary, and tertiary amines in which the usual UV–visible or fluorescent signal is complemented by paramagnetic signals of the same samples at different reporting times. The selectivity of the chemical probes for different types of amines and the large types of signals offered by these probes, now including paramagnetic species, will make them good starting points for the preparation of multi-signaling chemical probes for the detection of biogenic amines such as histamine, putrescine, and cadaverine. These amines were of particular interest due to their characteristics as markers for freshness in fish samples. To compare the different increases in fluorescence, the quantum yield of **1** was measured in DMSO in the presence of pyrrolidine, used as a reference, and then histamine, putrescine, or cadaverine, immediately after the addition of the amine. The reference was quinine sulfate in H_2_SO_4_ (0.05 M) (Figure 10). The results were very similar for all tested amines, with pyrrolidine being a bit higher than the rest of the amines within the error of the method (±0.02). In all cases, there was quite a noticeable fluorescence increase.

After testing in different conditions, the best conditions for the experiments were determined to be as follows: three mackerel (*Scomber scombrus*) samples were bought in a local market, the fish were cleaned, the spines were extracted, and the muscle was ground (Philips HR 28308/B). Then, fish samples of 10 ± 0.2 g were placed in 20 mL vials and sealed with a septum. Half of the samples were incubated at 4 °C, and the other half were incubated at 25 ± 1 °C. Then, 3 mL of the gas phase of the vial was taken with a syringe and bubbled into a probe solution at different times as the fish decayed over time. The absorbance and fluorescence of the samples were measured at their maximum peaks, and the differences between the obtained values and a reference solution were plotted (Figure 11).

The process was repeated three times and the results were combined. The results showed that the variation was quite low for refrigerated samples, but the emission of the samples left at 25 °C clearly increased over time. The results indicate that the changes in absorbance and fluorescence over time were closely related to the previous model amines. While the change in fluorescence or absorbance was not very significant in refrigerated samples, the amine concentration in the gas phase clearly increased over time for samples at 25 °C. The increase in fluorescence was more visible by eyesight under 366 nm light than by the naked eye (Figure 12).

Because it was easily visible under a UV light, the system could be used to create an intelligent label for the detection of amines. To do so, the probe was dissolved in DMSO, and a typical gelling agent (*N*-benzyloxycarbonyl-l-valyl-l-valine *n*-octadecylamide) [35] was added to the previous solution at 2 mg/0.5 mL DMSO, with 0.3% weight/volume being the minimum possible density to retain its gelation properties [36]. After that, a sample of biogenic amine from rotten fish was pumped with a syringe onto the probe (Appendix A), and a fading in the color and an increase in bright fluorescence were immediately apparent and lasted for a long time (Figure 13). The fluorescence decreased over time and disappeared after 12 h.

This process is potentially suitable for use as a smart label for the freshness of fish; for example, it can be used with suspected rotten fish without opening the packaged fish container (Figure 14).

Using the combined techniques developed in this process, not only fluorescence but also EPR could be used to characterize the freshness of fish while new applications of dual fluorescence-spin-label probes [37] or sensor arrays [38] are developed for practical use.

## 3. Materials and Methods

The reactions performed with air-sensitive reagents were conducted under dry nitrogen. The solvents were previously distilled under nitrogen over calcium hydride or sodium filaments. Melting points were determined in a Gallenkamp apparatus and are not corrected. Infrared spectra were registered with potassium bromide tablets using a Nicolet Impact 410 spectrometer (Madison, WI, USA). NMR spectra were recorded on Varian Mercury-300 (Palo Alto, CA, USA) and Varian Unity Inova-400 (Palo Alto, CA, USA) machines in DMSO-*d_6_* or CDCl_3_. Chemical shifts are reported in ppm with respect to residual solvent protons, and coupling constants (*J*_X-X’_) are reported in Hz. Mass spectra were taken on a Micromass AutoSpec machine (Milford, MA, USA) by electronic impact at 70 eV. Quantitative UV–visible measurements were performed with a Varian Cary 300 Bio UV spectrophotometer (Palo Alto, CA, USA) in 1 cm UV cells thermostated at 25 °C. Fluorescence spectra were recorded on a Hitachi F-7000 FL spectrofluorometer (Minato-ku, Tokyo, Japan) in the range 300–700 nm in 1 cm path length quartz cells thermostated at 25 °C. Solution X-band EPR spectra at room temperature were recorded by using flat cells on a Bruker EMX spectrometer (Billerica, MA, USA) equipped with a Bruker ER 036TM NMR-teslameter and an Agilent 53150A microwave frequency counter to fit the magnetic field and the frequency inside the measuring cavity. Simulation of the EPR spectra was performed by using the SimFonia program (Bruker, Billerica, MA, USA) and the public domain WinSim program (WinSim Inc., Richmond, TX, USA), while spectra and graphics were plotted with Kaleidagraph v3.5 software (Synergy Software, Reading, PA, USA).

*Synthesis of 1-(4-methoxyphenyl-3-[4-(1-(dicyanomethyleneindan-5-yl)phenyl]urea***1**. 5-(4-Aminophenyl)indan-1-one: Pd(PPh_3_)_4_ (54 mg, 0.05 mmol) was added under nitrogen to a solution of 5-bromoindanone (410 mg, 1.94 mmol) in a toluene/butanol/water 4:1:0.4 mixture (50 mL), and the mixture was stirred for 30 min at room temperature. Then, solid 4-aminobenzeneboronic pinacol ester (447 mg, 2.04 mmol) and Na_2_CO_3_ (1028 mg, 9.70 mmol) were added, and the mixture was heated under reflux for 16 h. Then, the solvent was evaporated under reduced pressure, and the residue was added to water (20 mL) and extracted with CH_2_Cl_2_ (5 × 40 mL). The combined organic extracts were dried (Na_2_SO_4_) and evaporated, and the residue was purified by flash chromatography (silica, 3 × 30 cm), CH_2_Cl_2_, to obtain 5-(4-aminophenyl)indan-1-one (353 mg, 82 %) as a pale yellow solid, mp 250–251 °C. IR (KBr, cm^−1^): 3429, 3335, 1678 (C=O), 1592, 1297, 1278, 811. ^1^H NMR (DMSO-*d_6_*, 400 MHz) δ: 7.71 (s, 1H, ArH), 7.60 (s, 2H, ArH), 7.48 (d, *J* = 8.6 Hz, 2H, ArH), 6.68 (d, *J* = 8.6 Hz, 2H, ArH), 5.44 (s, br exch, 2H, NH_2_), 3.10 (m, 2H, CH_2_), 2.62 (m, 2H, CH_2_). ^13^C NMR and DEPT (DMSO-*d_6_*, 400 MHz) δ: 205.44 (C=O), 156.14 (C-NH_2_), 149.53, 146.91, 134.05, 127.90 (CHar), 125.84, 124.60 (CHar), 123.25 (CHar), 122.79 (CHar), 114.13 (CHar), 36.10 (CH_2_), 25.44 (CH_2_). MS (EI) *m*/*z* (%): 224 (M^+^ + 1, 17), 223 (M^+^, 100), 195 (18), 167 (8), 152 (9), 139 (4), 117 (7), 96 (3). HRMS (EI): calcd. for C_15_H_13_NO: 223.0997 (M^+^); found: 223.0998. 1-Dicyanomethylene-5-(4-aminophenyl)indane: Malononitrile (200 mg, 3.03 mmol) and DABCO (120 mg, 1.07 mmol) were added under nitrogen to 5-(4-aminophenyl)indan-1-one (200 mg, 0.90 mmol) dissolved in toluene (50 mL). The mixture was heated under reflux for 8 h and then added to water (30 mL), stirred for 5 min, and extracted with dichloromethane (5 × 30 mL). The combined organic extracts were dried (Na_2_SO_4_), and the solvent was evaporated under reduced pressure. The crude product was purified by flash chromatography (silica, 3 × 30 cm), from CH_2_Cl_2_ to CH_2_Cl_2_:AcOEt (15:1), to obtain 1-dicyanomethylene-5-(4-aminophenyl)indene (59 mg, 95 %) as an orange solid, mp: >250 °C (decomp.), IR (KBr, cm^−1^): 3491, 3382, 2221 (CN), 1627, 1600, 1561 (Car-Car). ^1^H NMR (DMSO-*d_6_*, 300 MHz) δ: 8.15 (d, *J =* 9.0 Hz, 1H, ArH), 7.75 (m, 2H, 2 × ArH), 7.54 (d, *J =* 8.7 Hz, 2H, CH_2_), 6.66 (d, *J* = 8.7 Hz, 2H, 2 × ArH), 5.62 (s, br, exch, 2H, NH_2_), 3.23 (m, 2H, CH_2_), 3.11 (m, 2H, CH_2_). ^13^C NMR and DEPT (DMSO-*d_6_*, 75 MHz) δ: 179.83 (C=C(CN)_2_), 156.43, 150.35, 147.42, 132.53, 128.20 (CHar), 125.34 (CHar), 124.90 (CHar), 124.72, 121.69 (CHar), 114.36 (CN), 114.15 (CHar), 113.95 (CN), 69.86 (C(CN)_2_), 34.65 (CH_2_), 29.41 (CH_2_). MS (EI) *m*/*z* (%): 272 (M^+^ + 1, 20), 271 (M^+^, 100), 243 (3), 214 (2), 206 (1), 204 (2), 189 (2), 135 (2). HRMS (EI): calcd. for C_18_H_13_N_3_: 271.1109 (M^+^); found: 271.1108. 1-(4-methoxyphenyl-3-[4-(1-(dicyanomethyleneindan-5-yl)phenyl]urea 1: 1-Dicyanomethylene-5-(4-aminophenyl)indane (300 mg, 1.10 mmol) was added to 4,4′-diisocyanatodiphenylether (139 mg, 0.55 mmol) dissolved in CHCl_3_ (20 mL), and the mixture was stirred at room temperature for 48 h. Cyclohexane (4 mL) was then added, and the solid formed was filtered off, washed with CHCl_3_ (2 × 4 mL), and dried. Compound 1-(4-methoxyphenyl-3-[4-(1-(dicyanomethyleneindan-5-yl)phenyl]urea **1** (337 mg, 77%) was obtained as a dark yellow solid, mp: >250 °C. IR (KBr, cm^−1^): 3316, 2221 (C≡N), 1654 (C=O), 1598, 1557, 1510, 1239, 1184, 827. ^1^H NMR (DMSO-*d_6_*, 400 MHz) δ: 8.90 (s, 2H, 2 × NH), 8.73 (s, 2H, 2 × NH), 8.25 (d, *J =* 9.0 Hz, 2H, 2 × ArH), 7.91 (s, 2H, 2 × ArH), 7.87 (d, *J =* 9.0 Hz, 2H, 2 × ArH), 7.77 (d, *J =* 9.0 Hz, 4H, 4 × ArH), 7.61 (d, *J =* 9.0 Hz, 4H, 4 × ArH), 7.46 (d, *J =* 9.0 Hz, 4H, 4 × ArH), 6.96 (d, *J =* 9.0 Hz, 4H, 4 × ArH), 3.30 (m, 4H, 2 × CH_2_), 3.19 (m, 4H, 2 × CH_2_). ^13^C NMR and DEPT (DMSO-*d_6_*, 100 MHz) δ: 178.08 (C=C(CN)_2_), 156.27, 152.46, 151.95, 146.39, 140.92, 134.93, 133.73, 131.31, 127.81 (CHar), 125.95 (CHar), 125.40 (CHar), 123.26 (CHar), 120.11 (CHar), 118.82 (CHar), 118.41 (CHar), 114.09 (CN), 113.67 (CN), 71.23 (C(CN)_2_), 34.71 (CH_2_), 29.51 (CH_2_). MS (ESI) *m*/*z* (%): 795 (M^+^ + 1, 15). HRMS (ESI): calcd. for C_50_H_34_N_8_O_3_ + H^+^: 795.2827 (M^+^ + 1); found: 795.2832.

*Synthesis of 1,3-bis[4-(1-dicyanomethyleneindan-5-yl)phenylureidomethyl)benzene***2**. 1-Dicyanomethylene-5-(4-aminophenyl)indane (100 mg, 0.38 mmol) was added to *m*-xylylene diisocyanate (15 µL, 0.095 mmol) dissolved in CHCl_3_ (20 mL), and the mixture was stirred at room temperature for 24 h. Then, a second portion of m-xylylene diisocyanate (15 µL, 0.095 mmol) was added, and the mixture was stirred for an additional 24 h. The solid formed after this time was filtered off, washed with CHCl_3_ (2 × 2 mL), and dried under reduced pressure. Compound 1,3-bis[4-(1-dicyanomethyleneindan-5-yl)phenylureidomethyl)benzene **2** (104 mg, 75%) was obtained as a yellow solid, mp: 212–214 °C. IR (KBr, cm^−1^): 3328, 2221 (C≡N), 1658 (C=O), 1604, 1557, 1320, 1219, 1184. ^1^H NMR (DMSO-*d_6_*, 400 MHz) δ: 8.82 (s, 2H, NH), 8.17 (d, *J =* 9.0 Hz, 2H, 2 × ArH), 7.82 (s, 2H, 2 × ArH), 7.79 (d, *J =* 9.0 Hz, 2H, 2 × ArH), 7.65 (d, *J =* 8.6 Hz, 4H, 4 × ArH), 7.52 (d, *J =* 8.6 Hz, 4H, 4 × ArH), 7.29 (m, 2H, 2 × ArH), 7.20 (m, 2H, 2 × ArH), 6.71 (t, *J =* 6.0 Hz, 2H, 2 × NH), 4.32 (d, *J =* 6.0 Hz, 4H, 2 × CH_2_), 3.28 (m, 2H, CH_2_), 3.15 (m, 2H, CH_2_). ^13^C NMR and DEPT (CDCl_3_, 100 MHz) δ: 179.60 (C=C(CN)_2_), 155.78, 154.63, 146.04, 141.21, 140.00, 133.07, 130.10, 128.22 (CHar), 127.58 (CHar), 125.72, 125.67 (CHar), 125.55 (CHar), 125.22 (CHar), 122.60 (CHar), 117.54 (CHar), 113.66 (CN), 113.25 (CN), 70.61 (C(CN)_2_), 42.32 (2×CH_2_), 34.24 (CH_2_), 29.04 (CH_2_). MS (ESI) *m*/*z* (%): 753.27 (M^+^ + Na^+^, 94), 731.29 (M^+^ + 1, 100). HRMS (ESI): calcd. for C_46_H_34_N_8_O_2_ + H^+^: 731.2877 (M^+^ + 1); found: 731.2864.

*Synthesis of 1-(4-methoxyphenyl-3-[4-(1-(dicyanomethyleneindan-5-yl)phenyl]urea***3**. 1-Dicyanomethylene-5-(4-aminophenyl)indane (150 mg, 0.55 mmol) was added to *p*-methoxyphenyl isocyanate (82 mg, 0.55 mmol) dissolved in CHCl_3_ (10 mL), and the mixture was stirred at room temperature for 48 h. Cyclohexane (3 mL) was then added, and the solid formed was filtered off, washed with CHCl_3_ (2 × 2 mL), and dried. Compound 1-(4-methoxyphenyl-3-[4-(1-(dicyanomethyleneindan-5-yl)phenyl]urea **3** (172 mg, 74%) was obtained as a dark yellow solid, mp: 232–233 °C. IR (KBr, cm^−1^): 3320, 2221 (CN), 1658 (C=O), 1596, 1557, 1510, 1239, 1184, 1029, 823. ^1^H NMR (DMSO-*d_6_*, 300 MHz) δ: 8.85 (s, 1H, NH), 8.56 (s, 1H, NH), 8.23 (d, *J =* 9.0 Hz, 1H, ArH), 7.90 (s, 1H, ArH), 7.86 (d, *J =* 9.0 Hz, 1H, ArH), 7.76 (d, *J =* 9.0 Hz, 2H, ArH), 7.58 (d, *J* = 9.0 Hz, 2H, ArH), 7.37 (d, *J* = 9.0 Hz, 2H, ArH), 6.87 (d, *J =* 9.0 Hz, 2H, ArH), 3.82 (s, 3H, CH_3_), 3.28 (m, 2H, CH_2_), 3.18 (m, 2H, CH_2_). ^13^C NMR and DEPT (DMSO-*d_6_*, 75 MHz) δ: 180.04 (C=C(CN)_2_), 156.22, 154.52, 152.47, 146.33, 141.02, 133.63, 132.44, 131.05, 127.74 (CHar), 125.86 (CHar), 125.34 (CHar), 123.15 (CHar), 120.07 (CHar), 118.23 (CHar), 114.06 (CN), 113.94 (CHar), 113.63 (CN), 71.12 (C(CN)_2_), 55.13 (OCH_3_), 34.63(CH_2_), 29.45 (CH_2_). MS (EI) *m*/*z* (%): 421 (M^+^ + 1, 3), 420 (M^+^, 11), 297 (100), 271 (87), 149 (17), 123 (31). HRMS (EI): calcd. for C_26_H_20_N_4_O_2_: 420.1586 (M^+^); found: 420.1582.

*Synthesis of 1-[4-(N,N-dimethylamino)phenyl]-3-[4-(1-(dicyanomethyleneindan-5-yl)phenyl]urea***4**. 1-Dicyanomethylene-5-(4-aminophenyl)indane (150 mg, 0.55 mmol) was added to *p*-*N,N*-dimethylaminophenyl isocyanate (89 mg, 0.55 mmol) dissolved in CHCl_3_ (10 mL), and the mixture was stirred at room temperature for 48 h. Cyclohexane (3 mL) was then added, and the solid formed was filtered off, washed with CHCl_3_ (2 × 2 mL), and dried. Compound 1-[4-(*N,N*-dimethylamino)phenyl]-3-[4-(1-(dicyanomethylene-indan-5-yl)phenyl]urea **4** (160 mg, 67%) was obtained as a dark yellow solid, mp: 178–179 °C. IR (KBr, cm^−1^): 3320, 2221 (C≡N), 1654 (C=O), 1596, 1565, 1518, 1320, 1242, 1185, 823. ^1^H NMR (DMSO-*d_6_*, 300 MHz) δ: 8.81 (s, 1H, NH), 8.41 (s, 1H, NH), 8.25 (d, *J =* 9.0 Hz, 1H, ArH), 7.92 (s, 1H, ArH), 7.88 (d, *J =* 9.0 Hz, 1H, ArH), 7.76 (d, *J =* 9.0 Hz, 2H, 2 × ArH), 7.60 (d, *J =* 9.0 Hz, 2H, 2 × ArH), 7.29 (d, *J =* 9.0 Hz, 2H, 2 × ArH), 6.72 (d, *J =* 9.0 Hz, 2H, 2 × ArH), 3.30 (m, 2H, CH_2_), 3.20 (m, 2H, CH_2_), 2.84 (s, 6H, 2 × CH_3_). ^13^C NMR and DEPT (DMSO-*d_6_*, 75 MHz) δ: 180.06 (C=C(CN)2), 156.49, 152.63, 146.60, 141.28, 133.67, 130.96, 129.26, 127.80 (CHar), 125.89 (CHar), 125.42 (CHar), 123.16 (CHar), 120.32 (CHar), 118.22 (CHar), 117.54, 114.15 (CN), 113.73 (CN), 113.16 (CHar), 71.07 (C(CN)_2_), 40.64 (N(CH_3_)_2_), 34.61 (CH_2_), 29.43 (CH_2_). MS (EI) *m*/*z* (%): 433 (M^+^, 5), 297 (100), 271 (92), 162 (35), 136 (37). HRMS (EI): calcd. for C_27_H_23_N_5_O: 433.1903 (M^+^); found: 433.1907.

*Synthesis of 1-(4-methylthiophenyl)-3-[4-(1-(dicyanomethyleneindan-5-yl)phenyl]urea***5**. 1-Dicyanomethylene-5-(4-aminophenyl)indane (150 mg, 0.55 mmol) was added to *p*-methylthiophenyl isocyanate (91 mg, 0.55 mmol) dissolved in CHCl_3_ (10 mL), and the mixture was stirred at room temperature for 48 h. Cyclohexane (3 mL) was then added, and the solid formed was filtered off, washed with CHCl_3_ (2 × 2 mL), and dried. Compound 1-(4-methylthiophenyl)-3-[4-(1-(dicyanomethyleneindan-5-yl)-phenyl]urea **5** (173 mg, 72%) was obtained as a dark yellow solid, mp: 209–210 °C. IR (KBr, cm^−1^): 3317, 2221 (CN), 1654 (C=O), 1596, 1546, 1495, 1324, 1281, 1192, 827. ^1^H NMR (DMSO-*d_6_*, 400 MHz) δ: 8.91 (s,1H, NH), 8.77 (s,1H, NH), 8.25 (d, *J =* 9.0 Hz, 1H, ArH), 7.91 (s, 1H, ArH), 7.87 (d, *J =* 9.0 Hz, 1H, ArH), 7.77 (d, *J =* 9.0 Hz, 2H, ArH), 7.60 (d, *J =* 9.0 Hz, 2H, ArH), 7.44 (d, *J =* 9.0 Hz, 2H, ArH), 7.23 (d, *J =* 9.0 Hz, 2H, ArH), 3.30 (m, 2H, CH_2_), 3.19 (m, 2H, CH_2_), 2.44 (s, 3H, CH_3_). ^13^C NMR and DEPT (DMSO-*d_6_*, 75 MHz) δ: 180.05 (C=C(CN)_2_), 156.23, 152.28, 146.35, 140.80, 137.20, 133.72, 131.36, 130.17, 127.78 (CHar), 127.70 (CHar), 125.94 (CHar), 125.38 (CHar), 123.25 (CHar), 119.06 (CHar), 118.42 (CHar), 114.06 (CN), 113.64 (CN), 71.23 (C(CN)_2_), 34.69 (CH_2_), 29.50 (CH_2_), 15.94 (CH_3_). MS (EI) *m*/*z* (%): 436 (M^+^, 7), 297 (100), 271 (93), 165 (32). HRMS (EI): calcd. for C_26_H_20_N_4_OS: 436.1358 (M^+^); found: 436.1359.

*Synthesis of N,N-bis-[4-(1-(dicyanomethyleneindan-5-yl)phenyl]pyridine-2,6-dicarboxamide***6**. 2,6-Pyridinedicarbonyl dichloride (75 mg, 0.37 mmol) dissolved in dichloromethane (10 mL) was added dropwise to a chilled and stirred solution of 1-dicyanomethylene-5-(4-aminophenyl)indane (200 mg, 0.74 mmol) and *N,N*-diisopropylethylamine (96 mg, 0.74 mmol) in dichloromethane (50 mL). The mixture was left to warm to room temperature and stirred for 24 h. The solid formed was filtered off, washed with CHCl_3_ (2 × 3 mL), and dried. Compound *N,N*-bis-[4-(1-(dicyanomethyleneindan-5-yl)phenyl]pyridine-2,6-dicarboxamide **6** (197 mg, 79%) was obtained as a dark yellow solid, mp: >250 °C. IR (KBr, cm^−1^): 3344, 2221 (C≡N), 1682 (C=O), 1565, 1526, 1340, 1188, 819. 1H NMR (CDCl_3_, 400 MHz) δ (ppm): 10.79 (s, 2H, 2 × NH), 8.51 (d, *J =* 12.0 Hz, 2H, 2 × ArH), 8.42 (d, *J =* 12.0 Hz, 2H, 2 × ArH), 8.16 (d, *J =* 12.0 Hz, 2H, 2 × ArH), 8.10 (d, *J =* 9.0 Hz, 4H, 4 × ArH), 7.72 (m, 8H, 8 × ArH), 3.30 (m, 4H, 2 × CH_2_), 3.24 (m, 4H, 2 × CH_2_). ^13^C NMR and DEPT (DMSO-*d_6_*, 75 MHz) δ: 179.95 (C=C(CN)_2_), 161.74, 156.13, 148.69, 146.00, 139.99 (CHar), 139.00, 134.04, 133.80, 127.65 (CHar), 127.05, 126.20 (CHar), 125.35 (CHar), 123.61 (CHar), 121.09 (CHar), 113.93 (CN), 113.52 (CN), 71.50 (C(CN)_2_), 34.65 (CH_2_), 29.48 (CH_2_). MS (ESI) *m*/*z* (%): 674 (M^+^ + 1, 12). HRMS (EI): calcd. for C_43_H_27_N_7_O_2_ + H^+^: 674.2299 (M^+^); found: 674.2295.

*Synthesis of 5,5′-(((6-cyclohexyl-[3,4′-bipyridine]-2′,6′-dicarbonyl)bis(piperazine-4,1-diyl))bis(pyridine-6,3-diyl))bis(indan-1-one)***7**. 6-(Piperidin-1-yl)-[3,4′-bipyridine]-2′,6′-dicarboxylic acid: Pd(PPh_3_)_4_ (58 mg, 0.05 mmol) was added under nitrogen to a solution of dimethyl 4-bromopyridine-2,6-dicarboxylate (822 mg, 3.0 mmol) in tetrahydrofuran (120 mL), and the mixture was stirred for 30 min at room temperature. Then, solid 2-(piperidin-1-yl)-5-(4,4,5,5-tetramethyl-1,3,2-dioxaborolan-2-yl)pyridine (865 mg, 3.0 mmol) and cesium carbonate (1.47 g, 4.5 mmol) dissolved in water (15 mL) were successively added, and the mixture was heated under reflux for 24 h. Then, the solvent was evaporated under reduced pressure, and the residue was suspended in water (10 mL). Then, NaOH solution (5% *w*/*v* in H_2_O) was added dropwise until pH = 10, and the mixture was dissolved in tetrahydrofuran (60 mL) and heated under reflux for 1 h until complete hydrolysis of the diester and monoester mixture. Then, the mixture was chilled in an ice bath, and HCl solution (35% *w*/*v* in water) was added dropwise until pH = 1. Then, the solvent was evaporated until reduced pressure was achieved, and the solid residue was dried under vacuum. The solid residue was suspended in anhydrous dichloromethane (125 mL) under nitrogen, oxalyl chloride (3.00 g, 24 mmol) and anhydrous dimethylformamide (0.2 mL) were added, and the mixture was stirred overnight. Then, the solvent and the excess of oxalyl chloride were evaporated under reduced pressure, anhydrous methanol (5 mL, 124 mmol) in dichloromethane (60 mL) was added under nitrogen, and the mixture was stirred overnight. Then, the solvent was evaporated under reduced pressure, and the residue was purified by flash chromatography (silica, 3 × 50 cm), from CH_2_Cl_2_ to CH_2_Cl_2_/MeOH 9:1 *v*/*v*, to obtain dimethyl 6-(piperidin-1-yl)[3,4′-bipyridine]-2′,6′-dicarboxylate as a light yellow solid (522 mg, 49 %), mp: 146–148 °C, Rf (CH_2_Cl_2_/MeOH, 50:3): 0.83; ^1^H RMN (CDCl_3,_ 300 MHz): δ = 8.63 (s, 1H, ArH); 8.47 (s, 2H, ArH); 7.87–7.83 (dd, *J* = 9.3 Hz, 1H, ArH); 6.75–6.72 (d, *J* = 9.3 Hz, 1H, ArH); 4.04 (m, 6H, 3 × CH_2_); 3.66 (m, 4H, 3 × CH_2_); 1.68 (m, 9H, 3 × CH_2_); ^13^C RMN (CDCl_3_, 75 MHz): δ = 165.4 (C=O), 159.6 (C_Ar_), 148.7 (CH_Ar_), 148.6 (CH_Ar_), 147 (CH_Ar_), 135.5 (C_Ar_), 123.6 (CH_Ar_), 119.1 (C_Ar_), 106.5 (CH_Ar_), 53.4 (CH_3_), 45.9 (CH_2_), 25.5 (CH_2_), 24.7 (CH_2_); MS (EI) *m*/*z* (%): 355 (M^+^, 100), 326 (55), 312 (30), 299 (20), 272 (32), 84 (28); HRMS (EI): calcd. for C_19_H_21_N_3_O_4_: 355.1532; found 355.1535. The diester (512 mg, 1.44 mmol) was dissolved in a mixture of tetrahydrofuran (10 mL) and water (1 mL), a solution of NaOH (5% *w*/*v* in H_2_O) was added dropwise until pH = 10, and the mixture was heated under reflux for 1 h until complete hydrolysis of the diester. Then, the mixture was chilled in an ice bath, and HCl solution (35% *w*/*v* in water) was added dropwise until pH = 1. Then, the solvent was evaporated until reduced pressure was achieved, and the solid residue was extracted several times with dichloromethane/methanol 1:1 until complete extraction of the diacid. Evaporation of the solvent under reduced pressure afforded pure 6-(piperidin-1-yl)-[3,4′-bipyridine]-2′,6′-dicarboxylic acid as a light yellow solid (448 mg, 95% from the diester, 47% from dimethyl 4-bromopyridine-2,6-dicarboxylate), mp: 244–246 °C (decomp.); ^1^H RMN (CD_3_OD, 300MHz): δ = 8.49 (s, 2H, ArH); 8.43 (s, 0.3H, ArH); 8.28 (s, 0.7H, ArH); 8.14–8.11 (d, *J* = 9.3 Hz, 1H, ArH); 7.10–7.07 (d, *J* = 9.3 Hz, 1H, ArH); 3.66 (m, 4H, 3 × CH_2_); 1.69 (m, 6H, 3 × CH_2_); ^13^C RMN (CD_3_OD, 75 MHz): δ = 167.4 and 166.4 (C=O), 150.8 (C_Ar_), 149.9 (CH_Ar_), 140.2 (CH_Ar_), 131.2 (CH_Ar_), 128.8 (C_Ar_), 124.7 (CH_Ar_), 121.1 (C_Ar_), 111.4 (CH_Ar_), 26.6 (CH_2_), 25.2 (CH_2_); IR (KBr): 3609 (OH), 2786, 2167, 1648, 1340, 1108, 752 cm^−1^; MS (EI) *m*/*z* (%): 327 (M^+^, 75), 283 (92), 239 (45), 157 (41), 84 (100). HRMS (EI): calcd. for C_17_H_17_N_3_O_4_ + H^+^: 328.1292; found: 328.1291. 

5,5′-(((6-Cyclohexyl[3,4′-bipyridine]-2′,6′-dicarbonyl)bis(piperazine-4,1-diyl))bis(pyridine-6,3-diyl))bis(indan-1-one) **7**: Oxalyl chloride (530.0 μL, ρ = 1.478 g/mL, 6.17 mmol) was added to a suspension of 6-cyclohexyl-[3,4′-bipyridine]-2′,6′-dicarboxylic acid (294 mg, 0.90 mmol) in CH_2_Cl_2_ (100 mL) with three drops of DMF as a catalyst. The mixture was stirred at room temperature for 24 h, and then the solvent and the excess of oxalyl chloride were evaporated under reduced pressure. A solution of 1-dicyanomethylene-5-(4-aminophenyl)indane (190 mg, 0.70 mmol) in DMF (55 mL) was added over the white solid in an ice bath. Then, *N*,*N*-diisopropylethylamine (200 μL, ρ = 0.742 g/mL, 1.15 mmol) and a spatula tip of 4-dimethylaminopyridine (DMAP) were added, and the mixture was stirred at 0 °C for 15 min and then at room temperature for 24 h. A yellow precipitate formed that was filtered off. The filtered solid was washed with CHCl_3_ (3 × 10 mL) and CH_2_Cl_2_ (3 × 5 mL). Finally, the solid was collected, and traces of the solvent were evaporated under reduced pressure to obtain 5,5′-(((6-cyclohexyl-[3,4′-bipyridine]-2′,6′-dicarbonyl)bis(piperazine-4,1-diyl))bis(pyridine-6,3-diyl))bis(indan-1-one) **7** (466 mg, 62%) as an orange solid, almost insoluble in all common solvents, mp: 212-213 °C (decomp.); IR (KBr, cm^−1^): 3626 (NH), 3069, 2360, 2218, 1648 (C=O), 1558, 1527, 1337, 1188, 1122, 1005 cm^−1^; ^1^H NMR (DMSO-*d_6_*, 400 MHz) δ: 11.21 (s, 2H, 2 × NH), 8.72 (s, 1H, ArH), 8.61 (s, 2H, ArH), 8.31 (d, *J* = 8.4 Hz, 2H, ArH), 8.21 (m, 2H, ArH), 8.17 (d, *J* = 8.4 Hz, 4H, ArH), 8.00 (s, 2H, ArH), 7.95 (d, *J* = 8.8 Hz, 7H, ArH), 7.08 (d, *J* = 7.2 Hz, 1H, ArH), 3.70 (m, 4H, 2 × CH_2_)), 3.35 (m, 4H, 2 × CH_2_), 3.24 (m, 4H, 2 × CH_2_), 1.68 (m, 2H, CH_2_), 1.61 (m, 4H, 2 × CH_2_); MS (FAB) *m*/*z* (%): 833 (M^+^, 2), 834 (4), 835 (10), 836 (6), 837 (3). HRMS (EI): calcd. for C_53_H_39_N_9_O_2_ + H^+^: 834.3299; found: 834.3298.

## 4. Conclusions

In conclusion, we describe a rare class of persistent radicals formed through the interaction of fluorogenic arylurea derivatives and primary or secondary amines in the presence of light, water, and air. The most persistent species of the series was studied using spectroscopic techniques as well as quantum mechanics calculations. All results pointed to a rare case of photoinduced proton-coupled electron transfer, although the slow kinetics of formation and the evolution of the radical species from a formerly fluorescent transient intermediate rendered the system more complicated than classic examples. The two best examples of the formation of radicals having a well-resolved EPR hyperfine structure and high intensity came from a well-known selective fluorogenic probe for biogenic amines or a fluorescent probe for ω-amino acids, therefore adding a new perspective to these and other to-be-obtained fluorescent probes. These chemical probes are the first examples of multi-signal reporters for the discrimination between primary, secondary, and tertiary amines in which the usual UV–visible or fluorescent signal is complemented by paramagnetic signals of the same samples at different reporting times. The selectivity of the chemical probes for different types of amines and the large types of signals offered by these probes, now including paramagnetic species, make them good starting points for the preparation of multi-signaling chemical probes for the detection of important organic metabolites. As an immediate application, we developed a multi-signal smart label for the quick inspection of freshness in fish.

## Data Availability

The data presented in this study are contained within the article and also available in the Appendix A.

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
