# Peer review of "Synthesis of Fluorogenic Arylureas and Amides and Their Interaction with Amines: A Competition between Turn-on Fluorescence and Organic Radicals on the Way to a Smart Label for Fish Freshness"

_molecules, 2021, doi:10.3390/molecules26051404_

Round 1
Reviewer 1 Report
The manuscript by García-Tojal et al. reports an in-depth investigation of the synthesis, the spectroscopic properties and kinetic evolution of UV-vis absorbance, fluorescence and EPR signal of a family of fluorescent arylureas fluorescent probes for the detection of primary and secondary amines. Structurally related aryamides were also investigated for comparison. The investigation nicely clarifies the mechanism of fluorescence development and progressive disappearance for the investigated probe upon interaction with the amines, and brings to light for the first time the rare ability of such probes to act as dual reporter using fluorescence of the EPR signal, which was shown to build up on fluorescence fading. Although the best performance was obtained with two already know compounds of the family, the in-depts investigation of mechanism and kinetics significantly moves forward the knowledge and understanding of these compounds, opening to new rational developments in the field. Authors also developed and tested an assay for fish freshness based on said probes and designed a smart label for fish products to track freshness in the supply chain, without opening the primary packaging or subjecting the product to lab analysis.
The experimental procedures appear sound, the manuscript is clear and well written, despite the length, and the conclusions are sober and fully supported. Overall, I found the manuscript very interesting and I recommend publication in its current form.
Author Response
I, on behalf of all authors, want to thank the referee for his/her approval of the manuscript and for the valuable comments about the suitability of its publication.
Reviewer 2 Report
In this manuscript, Dr. Torroba and coworkers have synthesized a range of chemical probes, and they have studied their response through radical process. Furthermore, they have characterized potent intermediates by EPR spectrums. Overall, their study have provided a potential application of using those small molecules for detection of fish freshness. Thus, this manuscript should be accepted.
Author Response

(The authors gave the same response as above.)

Reviewer 3 Report
Manuscript Number: molecules-1124971
entitled: Synthesis of Fluorogenic Arylureas and Amides, and Their Interaction with Amines: A Competition Between Turn-on Fluorescence and Organic Radicals on the Way to a Smart Label for Fish Freshness
I had great pleasure reviewing this article. This is a well-conducted scientific study, done thoroughly and expressed concisely. Therefore, the manuscript is suitable for Molecules after considering the below comments:
- Please removed “dot” from the title.
- Did the authors compare presented EPR results with capto-dative stable radicals formed from thermal decomposition of azobisisobutyronitrile (abbreviated AIBN)? Both types of radicals possess some structural similarity.
- The Authors used secondary amines as “H-donors-Proton-traps (or sponge).” Did the Authors perform reactions with reagents, which could be “D-donors”? Labeling of a substance by isotope exchange reaction to proofed presented results. Deuterium incorporation into the product will be detected by EPR (for radical or paramagnetic), NMR, or MS, and it will be excellent evidence for the assumed presented mechanism (results).
- Because the Authors observed a change of color, in my opinion, it should be nice to present some differences in color in Fig. 9 (page 9) to distinguish some structures; they could correspond with actual colors.
- To Exp. Part. Please use only DCM or CH2Cl2 (see page 15).
Author Response
COMMENTS
- Please removed “dot” from the title.
- Did the authors compare presented EPR results with capto-dative stable radicals formed from thermal decomposition of azobisisobutyronitrile (abbreviated AIBN)? Both types of radicals possess some structural similarity.
- The Authors used secondary amines as “H-donors-Proton-traps (or sponge).” Did the Authors perform reactions with reagents, which could be “D-donors”? Labeling of a substance by isotope exchange reaction to proofed presented results. Deuterium incorporation into the product will be detected by EPR (for radical or paramagnetic), NMR, or MS, and it will be excellent evidence for the assumed presented mechanism (results).
- Because the Authors observed a change of color, in my opinion, it should be nice to present some differences in color in Fig. 9 (page 9) to distinguish some structures; they could correspond with actual colors.
- To Exp. Part. Please use only DCM or CH2Cl2 (see page 15).
RESPONSE
I, on behalf of all authors, want to thank the referee for his/her approval of the manuscript and for the valuable comments about the suitability of its publication.
1.-I have removed the dot from the title.
2.-Thanks for the valuable comments. I have added a short paragraph (page 4, lines, -1 to -3) and a few references [20-21] and [22-25] to stress the similarity of the reported radicals with the AIBN radicals and the likely capto-dative effect in the structures.
3.-Thanks for the valuable comments, we have not performed experiments to study the effect of deuterium on the reported radicals, at the moment we do not have the required deuterated intermediates (only deuterated solvents for NMR) and the preparations of them could take a somewhat long time, but I will keep the idea in mind because it will be a nice starting point for another subsequent paper.
4.-I have added color figures to Fig. 9.
5.-I have used CH2Cl2 along the paper.